# Totally X-ray-Free Ultrasound-Guided Mini-Percutaneous Nephrolithotomy in Galdakao-Modified Supine Valdivia Position: A Novel Combined Surgery

**DOI:** 10.3390/jcm11226644

**Published:** 2022-11-09

**Authors:** Yi-Yang Liu, Yen-Ta Chen, Hao-Lun Luo, Yuan-Chi Shen, Chien-Hsu Chen, Yao-Chi Chuang, Ko-Wei Huang, Hung-Jen Wang

**Affiliations:** 1Department of Urology, Kaohsiung Chang Gung Memorial Hospital and Chang Gung University College of Medicine, Kaohsiung 83301, Taiwan; 2Department of Electrical Engineering, National Kaohsiung University of Science and Technology, Kaohsiung 80778, Taiwan

**Keywords:** mini-PCNL, ultrasound guidance, GMSV position

## Abstract

We introduced a novel surgery that combines ultrasound guidance, miniaturization and Galdakao-modified supine Valdivia (GMSV) position in percutaneous nephrolithotomy (PCNL) and evaluated the safety and efficacy. This retrospective, single-center study retrospectively reviewed 150 patients who underwent ultrasound-guided mini-PCNL in the GMSV position from November 2019 to March 2022. All perioperative parameters were collected. Stone-free status was defined as no residual stones or clinically insignificant residual fragments (CIRF) <0.4 cm on postoperative day one. Among the 150 patients, the mean age was 56.96 years. The mean stone size was 3.19 cm (427 mm^2^). The mean S.T.O.N.E. score was 7.61, including 36 patients (24%) with scores ≥9. The mean operative time was 66.22 min, and the success rate of renal access creation in the first attempt was 88.7%. One hundred and forty (93.3%) patients were stone free. The mean decrease in Hemoglobin was 1.04 g/dL, and no patient needed a blood transfusion. Complications included transient hematuria (*n* = 13, 8.7%), bladder blood clot retention (*n* = 2, 1.3%), fever (*n* = 15, 10%) and sepsis (*n* = 2, 1.3%). Totally X-ray-free ultrasound-guided mini-PCNL in the GMSV position is feasible, safe and effective for patients with upper urinary tract stones, indicating the synergistic and complementary effects of the three novel techniques.

## 1. Introduction

Percutaneous nephrolithotomy (PCNL) was first introduced in 1976 [1], and over the years, it has become the gold standard of surgical treatment for renal stones larger than 2 cm [2]. Conventionally, PCNL was performed via a larger percutaneous nephrostomy (PCN) tract (≥22 French) [3], under fluoroscopic guidance, with patients in the prone position. Gradually, three novel techniques have been developed and widely accepted. First, ultrasound-guided PCNL reduces or even eliminates radiation exposure by fluoroscopy [4]. Moreover, mini-PCNL with miniaturization of the PCN tract (<22 French) [3] decreases renal trauma compared to standard PCNL [5]. Moreover, mini-PCNL demonstrated non-inferior surgical outcomes to standard PCNL for 2- to 4-cm-sized renal stones [6]. Finally, the Galdakao-modified supine Valdivia (GMSV) position facilitates simultaneous bidirectional endourological procedures rather than using the prone position [7].

Nevertheless, each of these three techniques also has its own weak points. First, it is not easy to monitor the process of PCN tract dilation using ultrasound guidance [8]. In addition, mini-PCNL is associated with lower lithotripsy efficiency and longer operative time [9]. Finally, the GMSV position may lead to renal displacement during PCN tract dilation and a narrow operating space during lithotripsy [10]. Fortunately, these three techniques have complementary advantages when they are combined. In the GMSV position, retrograde semi-rigid ureteroscopic assistance can be used to increase the safety of the puncture and dilation process [11]. In addition, the GMSV position improves the efficiency of mini-PCNL lithotripsy by the horizontal or downward axis of the Amplatz-type renal sheath [10]. Moreover, the GMSV position avoids repositioning from the lithotomy position to the prone position and, therefore, decreases the total operative time [12]. Mini-PCNL makes up for the insufficient operating space in the GMSV position [13]. Finally, ultrasound guidance facilitates PCNL in the supine position, including in the GMSV position [8].

Based on these complementary properties, we combined these three techniques for PCNL. To the best of our knowledge, studies of PCNL using the three combined techniques are limited. Therefore, we conducted a retrospective, single-center study to evaluate the outcomes of patients undergoing totally X-ray-free ultrasound-guided mini-PCNL in the GMSV position.

## 2. Materials and Methods

### 2.1. Study Design and Sample

This retrospective cohort study retrospectively reviewed the data of consecutive patients with upper urinary tract stone disease who had undergone one-step totally X-ray-free ultrasound-guided single-tract mini-PCNL in the GMSV position from November 2019 to March 2022 at Kaohsiung Chang Gung Memorial Hospital. Patients with age <18 years old, pregnancy, radiolucent stone, abnormal upper urinary tract anatomy (including horseshoe kidney, renal duplication, ureteropelvic junction obstruction, or ureteral stricture), preoperative severe urinary tract infection such as acute pyelonephritis or urosepsis, bleeding tendency, concurrent malignancy, multiple-tract PCNL, concurrent bilateral urinary tract endoscopic stone surgery, incomplete perioperative data or loss of follow-up were excluded. Finally, a total of 150 patients were included in the study.

### 2.2. Ethical Considerations

The protocol of the present study was approved by the Institutional Review Board of Kaohsiung Chang Gung Memorial Hospital (No. 202201106B0). Due to the retrospective study design, the IRB waived informed consent of the included patients.

### 2.3. Surgical Procedure and Statistical Analysis

All PCNL operations were performed by the same urologist (Dr. Yi Yang Liu). All included patients received basic preoperative examination, including non-contrast computed tomography (NCCT) of abdomen for image survey. The S.T.O.N.E. nephrolithotomy score (a graded system to predict patients’ stone-free status) was calculated according to NCCT findings [14]. Moreover, preoperative urinary culture was collected. If the result was positive, we would use intravenous antibiotics for the pathogen during the perioperative period. Otherwise, prophylactic antibiotics would be administered to the patients 30 min before the operation and kept for 24 h after the operation.

The patient was placed in the GMSV position under general anesthesia [7]. Ipsilateral 4 or 5 French ureteral catheterization was performed initially to create artificial hydronephrosis by manual ureteral catheter injection of 0.9% sodium chloride solution. Then, an ultrasound-guided (BK5000, BK Medical, Herlev, Denmark) 18-gauge needle transpapillary puncture toward the target renal calyx was performed with the assistance of a puncture frame. The needle tip in the renal collecting system was confirmed by the urine efflux from the puncture needle sheath, and the puncture depth was then measured. Subsequently, a 0.035-inch J-tip guidewire was indwelled into the puncture needle sheath, and a 0.6 cm skin incision was made. Sequentially, both 8 and 10 French fascial dilators were followed by the puncture depth. Finally, an 18 French Ultraxx^TM^ Nephrostomy Balloon Catheter (Cook Medical, Bloomington, IN, USA) was indwelled and inflated with 0.9% sodium chloride solution under the pressure of 20 atm for 3 min, and an 18 French Amplatz-type renal sheath was introduced to create the renal access. The dilation procedures were monitored by real-time ultrasound in as much detail as possible [15].

After creating the renal access, a 12 French Miniature Nephroscope (Richard Wolf, Knittlingen, Germany) and Holmium laser (Auriga XL 50 Watt, Boston Scientific, Boston, MA, USA) were used for stone fragmentation. The broken stone chips were washed out by low-pressure irrigation with 0.9% sodium chloride solution continuous irrigation from the mere height of 70 cm above the operating table. No irrigation pump or negative pressure suction device was used. Residual stones were checked by the nephroscope and ultrasound. Finally, a 4.7 or 6 French Double J stent was indwelled by the nephroscope. Either no catheter or a 14 French percutaneous nephrostomy balloon catheter was installed with 1 to 3 cc distilled water, depending on the surgeon’s decision. Simultaneously, retrograde semi-rigid ureteroscopy may be performed if indicated (e.g., failed artificial hydronephrosis creation by ureteral catheterization, confirmation of the guidewire or puncture needle tip in collecting system, residual stone in upper ureter or upper calyx, or failed antegrade Double J stenting). Operative time was defined as the time from ureteral catheterization to removal of the Amplatz sheath or the placement of the percutaneous nephrostomy balloon catheter.

Stone fragments were sent for analysis postoperatively. Blood examination and kidney ureter bladder (KUB) plain X-ray were performed on postoperative day one. Stone-free status was defined as no residual stone or clinically insignificant residual fragment (CIRF) <0.4 cm in KUB on postoperative day one. All perioperative data and events associated with postoperative surgical complications within one month were recorded. All descriptive statistics were analyzed using IBM SPSS version 21.0 Software (IBM, Armonk, NY, USA).

## 3. Results

The patients’ characteristics are listed in Table 1. Among the 150 patients, the mean age was 56.96 years, including 90 male patients and 60 female patients. Ninety-two patients underwent left-side PCNL. The mean body mass index (BMI) was 26 kg/m^2^, and 16.7% of the patients were obese (BMI > 30 kg/m^2^). The mean stone size and burden were 3.19 cm and 427 mm^2^, respectively. Twenty-two patients (14.7%) had staghorn stones, and 38 patients (25.3%) had both renal and upper ureteral stones. Mean stone density was 1199 Hounsfield units. Seventy percent of the patients have moderate to severe hydronephrosis. Twelve patients (8%) have history of percutaneous nephrolithotomy or open nephrolithotomy. High stone complexity (S.T.O.N.E. score ≧9) was noted in 36 patients (24%). The majority of patients (75.3%) belong to American Society of Anesthesiologists (ASA) classification 1 or 2. Preoperative mean hemoglobin (14.05 g/dL), mean creatinine (1.04 mg/dL), mean estimated glomerular filtration rate (eGFR) (74.3 mL/min/1.73 m^2^) and mean visual analog scale (VAS) for pain (0.35) were basically normal. In addition, 44 patients (29.3%) had positive urine cultures and underwent specific antibiotics treatment during the all-perioperative period.

Table 2 demonstrates the intraoperative parameters. The mean operative time was 66.22 min. Subcostal (93.3%) and middle calyceal (56.7%) punctures were used most frequently. The mean puncture depth was 8.84 cm. Thirty patients (20%) underwent non-hydronephrotic calyceal puncture with difficulty. However, the success rate of renal access creation on the first attempt was 88.7%. Retrograde semi-rigid ureteroscopic assistance was performed in 49 patients (32.7%). Tubeless procedures were performed in 21 patients (14%).

Postoperative outcomes are shown in Table 3. The mean hospital stay was 3.73 days, and immediate stone-free rate was 93.3% (140 patients). The mean reduction in hemoglobin was 1.04 g/dL. Compared to preoperative status, the mean postoperative eGFR was increased by 10.63 mL/min/1.73 m^2^. The mean postoperative VAS for pain was 2.99. Only 30 patients (20%) had postoperative VAS for pain ≥4 and needed postoperative intravenous analgesic agents for pain control. For stone analysis, 109 patients (72.7%) had calcium oxalate as the predominant stone. Regarding postoperative infection, 15 patients (10%) experienced fever >38 °C postoperatively. The fever was transient and subsided after antipyretic treatment in most patients. Only two patients (1.3%) had urosepsis but recovered soon without septic shock after broad-spectrum antibiotics treatment. In terms of hemorrhagic complications, 13 patients (8.7%) had transient gross hematuria that subsided spontaneously. Bladder blood clot retention was noted in two patients (1.3%) who underwent cystoscopic blood clot evacuation under general anesthesia. No blood transfusions, radiological interventions or nephrectomy for bleeding control were needed. Moreover, no intensive care unit transferation, chest or abdominal organ injury or mortality was noted. To sum up, the majority of the complications were classified as Clavien–Dindo Grade I. The incidence of Clavien–Dindo grade II and grade IIIb complications were only 1.3% and 1.3%, respectively.

## 4. Discussion

The results have revealed that totally X-ray-free ultrasound-guided mini-PCNL in the GMSV position is feasible with safety and efficacy. The mean operative time was about one hour, and the majority of cases had successful renal access creation on the first attempt. Postoperative outcomes showed that the majority of patients were stone free, and no major complication was noted. In the following discussion, we will analyze the detailed advantages through the whole process of PCNL. Figure 1 summarizes the three core techniques we used in the study and their effects on surgical outcomes.

The GMSV position, which is the combination of the oblique supine position and lithotomy position, simultaneously facilitates bidirectional endourological procedures without repositioning and saves significant operative time [7]. There is also no chest or abdominal compression in the GMSV position, which enables anesthesiologists to easily monitor and control each patient’s condition intraoperatively. Moreover, urologists can be seated with better ergonomics during the surgery [10]. Therefore, we can use the GMSV position throughout the procedures of PCNL with safety and efficacy.

In percutaneous renal calyceal puncture, ultrasound guidance requires no radiation exposure and provides easy identification of the posterior calyx and perirenal adjacent organs. In the present case series, no patient experienced pleura or perirenal organ injury. In addition, arterial puncture can be avoided under doppler mode ultrasound [8]. Hence, the risk of hemorrhagic complications is also decreased. The GMSV position also aids the puncture procedure because it allows retrograde semi-rigid ureteroscopic assistance to enhance retrograde ureteral irrigation when artificial hydronephrosis cannot be created by the ureteral catheter. Surgeons may also see the puncture needle tip or guidewire directly in the renal pelvis or ureter using the retrograde semi-rigid ureteroscope to ensure a successful renal puncture.

The rest part of renal access creation, including PCN tract dilation and Amplatz-type renal sheath setup, is a critical step before lithotripsy. Under the GMSV position, renal mobility is typically obvious because of the absence of abdominal compression, and it may lead to a shorter dilation or guidewire slippage and then failure of renal access creation [10]. To reduce renal mobility, we used skills such as coordinated abdominal counterpressure and brief apnea in maximal inspiration. Additionally, the use of the balloon dilator reduces the number of times of repetitive and sequential PCN tract dilation. Moreover, balloon dilation can be monitored under ultrasound during inflation [15]. Moreover, retrograde semi-rigid ureteroscopic assistance has been used for difficult cases by setting up a through-and-through guidewire to secure the subsequent renal access creation procedures [11]. In the present study, 20% of patients underwent non-hydronephrotic calyceal puncture. Even so, the success rate of renal access creation in the first attempt was still 88.7%. This result is comparable with that of another study in terms of ultrasound-guided conventional PCNL with balloon dilation in the prone position performed by very experienced urologists (88.4%) [15]. In other words, renal access creation by ultrasound-guided mini-PCNL in the GMSV position is shown to be feasible with a high success rate in the first attempt.

The vacuum cleaner effect of lithotripsy during mini-PCNL is the basic mechanism for stone fragment removal [16]. Conventionally, mini-PCNL often needs an irrigation pump with high irrigation pressure (150 to 250 mmHg) to effectively remove stone fragments [17]. In the GMSV position, the axis of the Amplatz-type renal sheath is horizontal or slightly inclined downward toward the ground. There is no doubt that this will enhance the vacuum cleaner effect compared to the prone position [10]. In the present study, just gravity irrigation with low irrigation pressure (70 cm H_2_O) was used for stone fragment removal, and there was no need for the irrigation pump. Additionally, compared to the standard PCNL, the mini-PCNL allows greater exploration from the single calyx to most of the desired locations in the renal collecting system without placing excessive torque on the renal parenchyma [13]. This advantage of the mini-PCNL compensates for the restricted working space and limited instrument movement through the longer PCN tract in the GMSV position [10]. Moreover, if residual fragments are found in the upper ureter or upper calyx, retrograde semi-rigid ureteroscopy is also readily available for lithotripsy. Although 24% of patients in the present study had complex renal stones with S.T.O.N.E. nephrolithotomy scores ≥9, the overall stone-free rate was still 93.3%, which was comparable with other studies of mini-PCNL (ranging from 75.0% to 95.1%) [18] or the pooled data from the latest meta-analysis (85.1%) [19].

In the literature review, Clavien–Dindo grade I to V complication rates of mini-PCNL were 2.7–20.8%, 1.4–17.3%, 0–10.3%, 0–0.05% and 0–0.02%, respectively [20]. The results of the current study were within the range and may prove the safety of our technique.

In addition to precise transpapillary renal puncture by ultrasound guidance, the miniaturization of the PCN tract is also associated with less renal trauma and lower bleeding risk and will lead to lower pain scale scores and fewer hemorrhagic complications [9,21]. In the present study, the mean decrease in hemoglobin is 1.04 g/dL. In addition, only two patients experienced bladder blood clot retention and underwent further cystoscopic blood clot evacuation. No patient needed a blood transfusion or radiological intervention for hemorrhage. Contemporary reports of mini-PCNL also showed a very low incidence of blood transfusion (<2%) [20]. Moreover, only 20% of the patients needed postoperative intravenous analgesics. These results indicated the minimal invasiveness of the procedure.

The incidence and severity of postoperative infection were low and acceptable in the present series. It is well known that mini-PCNL with a smaller Amplatz-type renal sheath wall causes higher intrarenal pressure, which leads to pyelovenous backflow [22] and has been identified as a risk factor for sepsis after PCNL [23]. However, the horizontal or downward axis of the Amplatz-type renal sheath in the GMSV position and low irrigation pressure by gravity rather than by irrigation pump decreases the intrarenal pressure significantly and helps to avoid postoperative infection [10]. In addition, the longer operative time is another risk factor for postoperative sepsis after PCNL [23]. However, in the GMSV position, the operative time was reduced not only by a single position throughout the whole procedure but also enhancement of the vacuum cleaner effect associated with the Amplatz-type renal sheath axis. Given the lower intrarenal pressure and shorter operative time in the present study, although 44 patients (29.3%) had positive preoperative urine cultures, only 15 patients (10%) experienced postoperative fever >38 °C, which was transient in most patients. Only two patients (1.3%) developed urosepsis, which was controlled by antibiotics administration. No patients developed septic shock. In the latest meta-analysis, the pooled incidence of fever after mini-PCNL is also about 10% [19]. Additionally, postoperative sepsis developed in 0.9–4.7% of patients after PCNL [20]. The results of current study were similar and acceptable.

The present study has several limitations. First, it was a retrospective, single-center study with the inherent limitations of these design factors. Moreover, it lacked a control group for comparison. The stone-free status was measured by KUB but not by computed tomography, which may lead to the under-detection of residual stone fragments. However, all surgeries were performed by the same urologist (Dr. Yi Yang Liu), which eliminates inter-surgeon bias. To the best of our knowledge, only a few studies have investigated PCNL in combination with the three novel techniques. Therefore, this study is a pioneer in exploring the combined PCNL techniques. Further prospective, multi-institutional comparative studies are still needed to confirm the safety and efficacy of this novel procedure compared to the conventional PCNL. Moreover, this combined technique may be suitable for some special situations, such as urolithiasis in solitary kidneys or transplant kidneys, to avoid severe intraoperative complications [24].

## 5. Conclusions

In this study, we found that ultrasound guidance, GMSV position and mini-PCNL are mutually complementary. Additionally, balloon dilation of the PCN tract and retrograde semi-rigid ureteroscopic assistance is helpful for renal access creation when performing ultrasound-guided PCNL under the GMSV position. Moreover, low-pressure gravity irrigation under the GMSV position ensures low intrarenal pressure and intraoperative safety. In conclusion, totally X-ray-free ultrasound-guided mini-PCNL in the GMSV position is feasible, safe and effective for patients with renal or upper ureteral stones, indicating the synergistic effects of the three novel techniques.

## Figures and Tables

**Figure 1 jcm-11-06644-f001:**
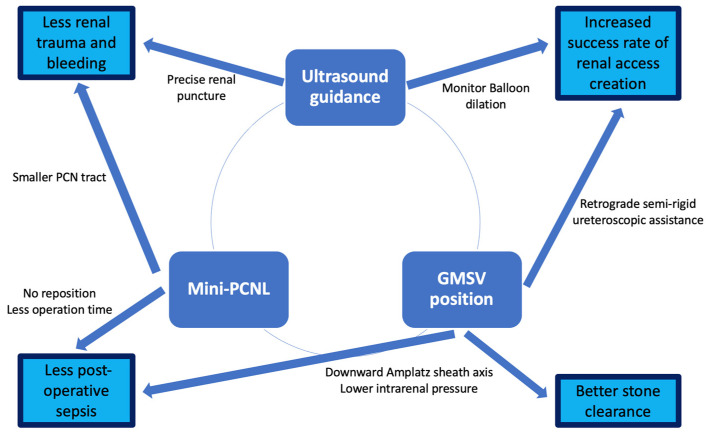
Three core techniques and their effects on surgical outcomes.

**Table 1 jcm-11-06644-t001:** Patients’ characteristics.

Variables		(*n* = 150)
Age, years (mean ± SD)		56.96 ± 12.45
Gender (Male/Female)		90/60
Laterality (Left/Right)		92/58
Body mass index, kg/m^2^ (mean ± SD)		26.00 ± 4.27
Obesity (Body mass index > 30 kg/m^2^), *n* (%)		25 (16.7%)
Total stone size, cm (mean ± SD)		3.19 ± 1.67
Total stone burden, mm^2^ (mean ± SD)		427 ± 360
Stone number		
	Single, *n* (%)	42 (28.0%)
	Multiple, *n* (%)	86 (57.3%)
	Staghorn stone, *n* (%)	22 (14.7%)
Stone location		
	Kidney, *n* (%)	90 (60.0%)
	Upper ureter, *n* (%)	22 (14.7%)
	Kidney and upper ureter, *n* (%)	38 (25.3%)
Stone density, Hounsfield unit (mean ± SD)		1199.1 ± 309.2
S.T.O.N.E. Score (mean ± SD)		7.61 ± 1.36
S.T.O.N.E. Score ≥ 9, *n* (%)		36 (24.0%)
Preoperative hydronephrosis		
	None, *n* (%)	32 (21.3%)
	Mild, *n* (%)	13 (8.7%)
	Moderate, *n* (%)	78 (52.0%)
	Severe, *n* (%)	27 (18.0%)
Previous surgery		
	ESWL or URSM or RIRS, *n* (%)	57 (38.0%)
	PCNL or open surgery, *n* (%)	12 (8.0%)
ASA classification	1, *n* (%)	5 (3.3%)
	2, *n* (%)	108 (72.0%)
	3, *n* (%)	36 (24.0%)
	4, *n* (%)	1 (0.7%)
Preoperative Hemoglobin, g/dL (mean ± SD)		14.05 ± 1.64
Preoperative Creatinine, mg/dL (mean ± SD)		1.04 ± 0.41
Preoperative eGFR (MDRD), mL/min/1.73 m^2^ (mean ± SD)		74.30 ± 24.23
Preoperative positive urine culture, *n* (%)		44 (29.3%)
Preoperative pain scale, visual analog scale (mean ± SD)		0.35 ± 0.83

SD = standard deviation; ESWL = extracorporeal shock wave lithotripsy; URSM = ureteroscopic stone manipulation; RIRS = retrograde intrarenal surgery; PCNL = percutaneous nephrolithotomy; ASA = American Society of Anesthesiologists; eGFR = estimated glomerular filtration rate; MDRD = modification of diet in renal disease.

**Table 2 jcm-11-06644-t002:** Intraoperative parameters.

Parameters		(*n* = 150)
Operative time, min (mean ± SD)		66.22 ± 36.54
Target calyx		
	Upper, *n* (%)	14 (9.3%)
	Middle, *n* (%)	85 (56.7%)
	Lower, *n* (%)	51 (34.0%)
Puncture site		
	11th intercostal space, *n* (%)	10 (6.7%)
	Subcostal area, *n* (%)	140 (93.3%)
Non-hydronephrotic calyceal puncture, *n* (%)		30 (20.0%)
Success of renal access creation in the first attempt, *n* (%)		133 (88.7%)
Puncture depth, cm (mean ± SD)		8.84 ± 1.90
Retrograde semi-rigid ureteroscopic assistance, *n* (%)		49 (32.7%)
Tubeless, *n* (%)		21 (14.0%)

SD = standard deviation.

**Table 3 jcm-11-06644-t003:** Postoperative outcomes.

Variables		(*n* = 150)
Hospital stay, days (mean ± SD)		3.73 ± 1.59
Stone-free status, *n* (%)		140 (93.3%)
Postoperative Hemoglobin, g/dL (mean ± SD)		13.01 ± 1.70
Hemoglobin drop, g/dL (mean ± SD)		1.04 ± 1.10
Postoperative Creatinine, mg/dL (mean ± SD)		0.92 ± 0.34
Postoperative eGFR (MDRD), mL/min/1.73 m^2^ (mean ± SD)		85.26 ± 27.47
Change of eGFR (MDRD), mL/min/1.73 m^2^ (mean ± SD)		10.63 ± 18.27
Postoperative pain scale, visual analog scale (mean ± SD)		2.99 ± 1.50
Stone analysis		
	Calcium oxalate predominant, *n* (%)	109 (72.7%)
	Calcium phosphate predominant, *n* (%)	41 (27.3%)
Complications classified by Clavien–Dindo classification		
Grade I		
	Fever > 38 °C, *n* (%)	15 (10.0%)
	Transient gross hematuria, *n* (%)	13 (8.7%)
	Postoperative pain scale ≥ 4, *n* (%)	30 (20.0%)
Grade II		
	Sepsis, *n* (%)	2 (1.3%)
Grade IIIb		
	Bladder blood clot retention, *n* (%)	2 (1.3%)

SD = standard deviation; eGFR = estimated glomerular filtration rate; MDRD = modification of diet in renal disease.

## Data Availability

The datasets generated during the current study are available from the corresponding author upon reasonable request.

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
