# Peer review of "Totally X-ray-Free Ultrasound-Guided Mini-Percutaneous Nephrolithotomy in Galdakao-Modified Supine Valdivia Position: A Novel Combined Surgery"

_jcm, 2022, doi:10.3390/jcm11226644_

Round 1

Reviewer 1 Report

This is the report of a novel (according to the writers) technique for stone surgery. Overall the paper is well written and documented despite the fact that I think that a similar technique has already been published. Nevertheless I think that this report must be published without any major revisions

Author Response

Response to Reviewer 1 Comments

Point 1:

This is the report of a novel (according to the writers) technique for stone surgery. Overall the paper is well written and documented despite the fact that I think that a similar technique has already been published. Nevertheless I think that this report must be published without any major revisions

Response 1:

Thank you for your precious comments on our study.

The purpose of this study is to introduce this combined technique and arouse some urologists’ interest.

We highly recommend all the urologists to use this technique for their patients.

Thank you again for your agreement for publication.

Reviewer 2 Report

In the manuscript titled “Totally X-ray-free ultrasound-guided mini-percutaneous nephrolithotomy in Galdakao-modified supine Valdivia position: A novel combined surgery” Authors Yi Yang Liu et al. analyzed the safety and efficacy and surgical outcomes of mini-percutaneous nephrolithotomy carried out without the use of fluoroscopy.

This is a retrospective monocenter cohort study that include 150 patients surgically treated for stone disease with an interesting surgical approach that is currently spreading across several stone center. All the patients were treated in Galdakao-modified supine Valdivia position within the assistance of only ultrasound guidance for percutaneous access. All the procedure were conducted with a 12 French Miniature Nephroscope and Holmium laser without the use of irrigation pomp of negative pressure for aspiration of stone fragments. Unfortunately no control group is available.

Of note, ureteral catheterization for artificial hydronephrosis creation was performed before renal punction and retrograde semirigid ureteroscopy was performed in some cases (failed artificial hydronephrosis creation by ureteral catheterization, confirmation of the guidewire or puncture needle tip in collecting system, residual stone in upper ureter or upper calyx, or failed antegrade Double J stenting).

Stone free status was Stone-free status was defined as no residual stone or residual fragment < 0.4 cm in KUB on postoperative day one. 

Interestingly 16% of patient were obese, a condition that normally make the renal punction challenging notably without the fluoroscopy assistance. 

The success rate of renal access was 88.7% and semirigid ureteroscopy was performed in 1 patient out of 3. More that 90% of patient were stone free. 

Surprisingly, Authors report a very low rate of post-operative complications with just 2 cases of urinary sepsis and 2 case of blood clot urinary retention. Apparently, only 15 patients had post-operative fever that was spontaneously resolved and just 20% of patients required i.v. analgesics. In my opinion this rate seems to be underestimate. 

The manuscript is well written but does not demonstrate any new surgical technique as far as xray free miniperc in GMSV position are deeply and fully descripted in several previous articles. Moreover, this paper lacks a well conducted complications survey because are presented just complication occurred during hospitalization. In the same way the rate of 93% stone free patients seems to be overestimates. 

My comments to the Editors and Authors:

-       It could be interesting to have a short/medium-term complication and stone free status assessment at 1 week and 1 and 3 month post-surgery

-       Stone free status assessment should be evaualted within a CT scan in medium term follow up

-       Complication should be assessed with the clavien dindo classification

-       Use of post-operative antibiotics should be reported

-       Complication and stone free status should be deeply discussed with available literature

Another concern is about the use of semirigid ureteroscopy during the miniperc procedure (1 patient out of 3).

-       authors should deeply describe why and when resort to retrograde approach; is better to perform a semirigid ureteroscopy rather that xray for ultrasound guided difficult renal punction?  

-       Have authors performed combined ant. and retrograde lithotripsy?

-       Retrograde ureteral catheterization and semirigid ureteroscopy were performed without the use of xray control too?

-       In my opinion figure 1 does not result very clear or useful and should be revised

-       Is the operative time of 1 hour the lithotripsy time or the whole procedure time? It is hard to imagine that in just 1 hour surgeons can perform retrograde approach and that renal punction and dilatation and a complete lithotripsy

I thank the Editor for the opportunity of revise this manuscript.

Author Response

Response to Reviewer 2 Comments

Point 1:

-     It could be interesting to have a short/medium-term complication and stone free status assessment at 1 week and 1 and 3 month post-surgery

Response 1:

Thank you for your precious suggestion.

The purpose of our study is to investigate the perioperative outcomes of our technique. In other words, we want to see the direct effect of the surgery. Besides, because of retrospective design, the postoperative follow up protocol was not uniform and may lead to bias of the middle term outcome. Moreover, some auxiliary procedures may be taken and influence the stone free rate and complication rate. Therefore, we just evaluated the initial results including immediate stone free rate and surgical complication within one month, which were related to the operation directly.

Point 2:

-       Stone free status assessment should be evaualted within a CT scan in medium term follow up

Response 2:

Thank you again for the wonderful suggestion.

We have declared this limitation in the discussion. In fact, under the system of national health insurance in Taiwan, we cannot arrange CT scan routinely for all the patients as postoperative follow up because of the limited insurance budget. It is really a pity.

Point 3:

-       Complication should be assessed with the clavien dindo classification

Response 3:

We have added the Clavien-Dindo classification for complication classification.

Please refer to Table 3 of the revised manuscript for the details.

Point 4:

-       Use of post-operative antibiotics should be reported

Response 4:

We have added the principles of perioperative antibiotics to material and methods section.

Please refer to line 87 of the revised manuscript for the details.

Point 5:

-       Complication and stone free status should be deeply discussed with available literature

Response 5:

We have added the data of the reviewed literature in discussion.

Basically, our results were comparable with the literature for stone free status and complication.

Please refer to line 242, 247, 257, 277 of the revised manuscript for the details.

Point 6:

Another concern is about the use of semirigid ureteroscopy during the miniperc procedure (1 patient out of 3).

-       authors should deeply describe why and when resort to retrograde approach; is better to perform a semirigid ureteroscopy rather that xray for ultrasound guided difficult renal puncture?  

Response 6:

Thank you again for noticing one of the key points of our technique

We have described the indication of retrograde semi-rigid ureterosocopy in the materials and methods section. Please refer to line 117 of the revised manuscript for the details

We didn’t know which technique is better for ultrasound guided difficult renal puncture. But our technique is really useful and practical without radiation exposure.

Point 7:

-       Have authors performed combined ant. and retrograde lithotripsy?

Response 7:

Yes, we will do combined antegrade and retrograde lithotripsy if needed.

Point 8:

-       Retrograde ureteral catheterization and semirigid ureteroscopy were performed without the use of xray control too?

Response 8:

Yes, we didn’t use any X ray device during the whole operation procedures.

Point 9:

-       In my opinion figure 1 does not result very clear or useful and should be revised

Response 9:

We have revised Figure 1 and we hope it will become more comprehensive.

Please refer to Figue 1 of the revised manuscript for the details.

Point 10:

-       Is the operative time of 1 hour the lithotripsy time or the whole procedure time? It is hard to imagine that in just 1 hour surgeons can perform retrograde approach and that renal punction and dilatation and a complete lithotripsy

Response 10:

Thank you for the key question.

We have added the definition of our operative time.

Please refer to line 121 of the revised manuscript for the details.

Our operative time is the time of whole procedure.

In our opinion, no need for reposition and relatively small stone size (3.19 cm) would be the factors for less operative time.

Finally, we thank you for your efforts in reviewing this article.

Best regards.

Reviewer 3 Report

well written good focused a study on ultrasound-guided mini-percutaneous nephrolithotomy in Galdakao-modified supine Valdivia position.

Were there any kidney transplant patients among the patients in the study group?

Have the uthors used this method any solitery kidney patient with nephrolithiasis?

The authors can add the study below as a references to increase the value of content in discussion section

Authos can mention more about this approach such cases that I asked above in discussion section.

The authors can add the studies below as a references to increase the value of content in discussion section

Sarier M, Duman I, Yuksel Y, et al. Results of minimally invasive surgical treatment of allograft lithiasis in live-donor renal transplant recipients: a single-center experience of 3758 renal transplantations. Urolithiasis. February 2018. doi:10.1007/s00240-018-1051-0

Author Response

Response to Reviewer 3 Comments

well written good focused a study on ultrasound-guided mini-percutaneous nephrolithotomy in Galdakao-modified supine Valdivia position.

Point 1:

Were there any kidney transplant patients among the patients in the study group?

Response 1:

No, there were no kidney transplant patients in our study group.

Point 2:

Have the authors used this method any solitary kidney patient with nephrolithiasis?

Response 2:

No, we have not used this method for solitary kidney patients with nephrolithiasis.

Point 3:

Authors can mention more about this approach such cases that I asked above in discussion section.

The authors can add the studies below as a references to increase the value of content in discussion section

Sarier M, Duman I, Yuksel Y, et al. Results of minimally invasive surgical treatment of allograft lithiasis in live-donor renal transplant recipients: a single-center experience of 3758 renal transplantations. Urolithiasis. February 2018. doi:10.1007/s00240-018-1051-0

Response 3:

Thank you for your precious comment.

We have added this part to our future work in the discussion section and cite the article as reference.

Please refer to line 291 and reference No.24 of the revised manuscript for the details.

We will try to apply this technique to the patients you mentioned in the future.

Finally, we thank you for your efforts in reviewing this article.

Best regards.

Round 2

Reviewer 2 Report

Authors revised the manuscript according to my suggestions when needed and kindly answered my concerns.

Thanks

Reviewer 3 Report

none